# Treatment with Methylphenidate Improves Affective but Not Cognitive Empathy in Youths with Attention-Deficit/Hyperactivity Disorder

**DOI:** 10.3390/children8070596

**Published:** 2021-07-14

**Authors:** Pamela Fantozzi, Pietro Muratori, Maria Celeste Caponi, Valentina Levantini, Cristina Nardoni, Chiara Pfanner, Federica Ricci, Gianluca Sesso, Annalisa Tacchi, Annarita Milone, Gabriele Masi

**Affiliations:** 1IRCCS Stella Maris Foundation, Scientific Institute of Child Neurology and Psychiatry, Calambrone, 56128 Pisa, Italy; pietro.muratori@fsm.unipi.it (P.M.); celeste_caponi@hotmail.com (M.C.C.); valentina.levantini@fsm.unipi.it (V.L.); cristina.nardoni@yahoo.it (C.N.); chiara.pfanner@fsm.unipi.it (C.P.); federica.ricci@fsm.unipi.it (F.R.); gianluca.sesso@fsm.unipi.it (G.S.); annalisa.tacchi@fsm.unipi.it (A.T.); annarita.milone@fsm.unipi.it (A.M.); gabriele.masi@fsm.unipi.it (G.M.); 2Department of Clinical and Experimental Medicine, University of Pisa, 56126 Pisa, Italy

**Keywords:** attention deficit hyperactivity disorder, social impairment, cognitive empathy, affective empathy, methylphenidate

## Abstract

Background: Beside the core symptoms, patients with attention-deficit/hyperactivity disorder (ADHD) frequently show relevant difficulty in developing relationships with peers. Although ADHD symptoms may account for social impairment, deficits in cognitive and/or affective empathy have also been involved. Our aim was to investigate the effect of methylphenidate (MPH) treatment on affective and cognitive empathy. Methods: Sixty-one drug-naïve youths with ADHD (age range 6 to 17 years, mean 10.3 ± 2.8 years, 51 males) naturalistically treated with MPH monotherapy were followed up for 6 months for ADHD symptoms and empathy, measured with the Basic Empathy Scale. Results: After being treated with MPH, the patients showed a significant improvement in affective and cognitive empathy scores. Linear regression models showed that changes in inattention symptoms predicted changes in affective but not in cognitive empathy, while changes in the hyperactivity/impulsivity symptoms did not predict changes in affective or cognitive empathy. Conclusions: Our study provides a further contribution for a better understanding of the possible effects of the MPH on youth’s characteristics.

## 1. Introduction

Attention-deficit/hyperactivity disorder (ADHD) is a neurodevelopmental disorder with onset before 12 years of age, with a worldwide prevalence of around 7%, characterized by persistent inattention and/or hyperactivity/impulsivity, with significant social and academic impairment [1]. Patients with ADHD frequently show difficult relationships with peers [2]. Santosh and Mijovic (2004) [3] described two types of social impairment, relationship difficulties, related to the consequences of ADHD symptoms, and social communication difficulties, with symptoms in the autism spectrum disorder (ASD) domain.

ADHD symptoms, per se, may interfere with adequate social interactions. Disinhibition of motor, verbal and behavioral decisions can lead to fewer opportunities for social interaction due to peer rejection. Impulsivity involves inappropriately intruding upon conversations or play [4], while inattention interferes with the learning of social information, i.e., focusing and sustaining attention during conversations or appropriately reading social cues during play [5].

Regarding the overlap between ADHD and ASD symptoms, until DSM-5 [1], the possibility of a diagnosis of ADHD in autistic children was excluded, albeit extended epidemiological research showed that ASD and ADHD are commonly co-occurring. Up to 22–83% of children with ASD have symptoms fitting DSM criteria for ADHD, and 30–65% of children with ADHD have clinically significant symptoms of ASD [6,7]. Clinical implications of this co-occurrence are still under exploration [8]. ASD, along with other conditions such as psychopathy, have been described as “empathy disorders” [9]. A wide range of studies have reported a dissociation between the cognitive and the affective components of empathy in ASD [10,11,12,13,14]. The role of empathy in determining impaired social interactions in ADHD youths has received specific attention within the wider concept of social cognition deficits [2]. Empathy is a complex multidimensional construct, which includes affective empathy (AE), that is, the capacity to share emotions and respond to emotional displays of others, and cognitive empathy (CE), that is, the ability to understand the perspective of another person [15]. These two components may have different neuroanatomical correlates and it has been demonstrated that one process may be intact while the other is impaired [16].

Children with ADHD showed impaired AE, compared to normal controls [17,18,19], either assessed as a trait using parent reports [20] or state assessed with affective responses to vignettes [17]. Furthermore, the difficulties in emotion recognition, involved in AE [16], have been reported in youths with ADHD [21]. Maoz et al. (2019) [22] showed a global deficit in both the components of empathy in its self-report form. In another study from the same research group [23], differences in the empathic profile were identified between the combined (ADHD-C) and the inattentive (ADHD-I) subtypes of ADHD, with greater impairment in the first.

Attention is a key component of empathic responsiveness in normal developmental processes [24]. Inattentive children could present deficits in the comprehension of the valence of the social cues, peer intent and social outcomes because they simply do not notice these cues when they are presented to them [5,25].

Psychostimulants, including methylphenidate (MPH) and amphetamines, the gold-standard for the pharmacological treatment of ADHD [26,27], have been associated with improvements in both AE and CE in children and adolescents with ADHD. Some studies examined the effect of a single-dose administration of MPH on patients who were already regularly taking the medication at the time of the study [22,23,28], other studies evaluated the effect of mid-term treatment with daily administration of MPH [29,30,31]. Interestingly, MPH administration has been shown to promote empathy-like behaviors and sociability and reduce aggressiveness in a mouse model of callousness [32].

The aim of our study was to investigate the effect of MPH on AE and CE in a group of youths with a primary diagnosis of ADHD, all drug-naïve, after 6 months treatment. We assumed that the MPH treatment would be associated with an improvement in the children’s attention abilities and that this improvement would be associated with an improvement of both the components of empathy. Considering that ASD is a neurodevelopmental disorder often associated with a lack of empathy, we decided to exclude patients with the ASD comorbidity in this pilot study.

## 2. Materials and Methods

### 2.1. Participants

Sixty-six subjects with ADHD as main reason for referral participated in the present study, five of which with ASD comorbidity. The final clinical group included 61 drug-naïve youths aged between 6 and 17 years (mean age 10.3 ± 2.8 years, 51 males and 10 females) with ADHD, 50 with ADHD-C (82%) and 11 with ADHD-I (18%), recruited in our third-level Department of Child and Adolescent Psychiatry from January 2018 to April 2019. None of the patients were undergoing psychological or educational treatment at the time of the recruitment and they all attended school regularly. The comorbidities were: specific learning disorders (*N* = 14; 23%), oppositional defiant disorder (*N* = 9; 14.8%), mood disorders (*N* = 4; 6.6%), language disorder (*N* = 2; 3.3%), anxiety disorders (*N* = 1; 1.6%), tics (*N* = 1; 1.6%), and verbal dyspraxia (*N* = 1; 1.6%). All the subjects were diagnosed according to the DSM-5 criteria [1], based on clinical history and a structured interview, Kiddie Schedule for Affective Disorders and Schizophrenia–Present and Lifetime version [33]. The two patients with language disorder and the patient with verbal dyspraxia had previously received these diagnoses in the Neurology section of our Department. The ASD diagnosis were given by experienced clinicians according to the criteria of the DSM-5 [1] and it was made with Autism Diagnostic Observation Schedule, Second Edition [34]. The inclusion criteria were: (1) main diagnosis of ADHD; (2) a full scale IQ (WISC-IV [35]) of 80 or above; (3) caregivers consent to pharmacological treatment. Exclusion criteria were (1) autism spectrum disorder diagnosis and other neurological conditions, (2) an estimated full scale IQ < 80, and (3) any pharmacological treatment at the baseline.

### 2.2. Procedures

All the 61 patients were treated in monotherapy with MPH during the follow-up. At the baseline, they received a dose-test of MPH (5 or 10 mg, according to age and weight). After one week, the starting dose of MPH was increased, with subsequent titrations of 5–10 mg no more frequently than at 5-day intervals, with flexible titration, depending on age and weight, clinical outcome, and occurrence of side effects, based on weekly monitoring visits during the first month, then monthly. The final MPH dosage was 31.6 ± 15.1 mg/day (dose range 5 to 60 mg/day). All the patients were followed-up for 6 months. At baseline and at endpoint, parents completed the ADHD-Rating Scale-IV [36], and youths completed the Basic Empathy Scale [37].

### 2.3. Measures

*Categorical diagnosis*: The K-SADS-PL [33] was used to assess for current and past DSM-5 disorders. Clinicians conducting the interviews were trained and satisfied reliability criteria (*k* Cohen ≥ 0.80). Both parents and children participating in the study completed the K-SADS interview independently. The rate of child−parent K-SADS diagnosis agreement was 0.87 (*k* Cohen).

*Children’s Intelligence Quotient*: Cognitive abilities were assessed with the Wechsler Intelligence Scales for Children—4th Ed. [35].

*Cognitive and affective empathy:* The Basic Empathy Scale (BES) [37] is a self-reported questionnaire developed for children and adolescents, with 20 items across two subscales referring to AE and CE (0.79, Cronbach’s α in this sample; 0.82, Cronbach’s α in this sample). Higher scores indicate more empathy capacity.

*ADHD severity*: The ADHD-Rating Scale-IV (ADHD-RS) [36] is an 18-item questionnaire consisting of two subscales, Inattention (9 items) and Hyperactivity-Impulsivity (9 items), completed by the parents, that measures ADHD symptoms according to the DSM-5. Higher scores indicate the presence of more symptoms.

### 2.4. Statistical Analysis

All the statistical tests were run on IBM SPSS Statistics 23. The first step of the statistical analysis plan involved missing data handling. In total, 9.32% of the values were incomplete. Missing data were imputed using multiple imputations. Incomplete variables were imputed under fully conditional specification, using the default setting “Impute Missing Data Values (Multiple Imputation)” available on SPSS 23. As a preliminary analysis, we computed descriptive statistics and bivariate correlations of the study variables. The second step was to calculate a post hoc power analysis using the * Power 3.1.9 [38]; results of this analysis indicated a power >0.90 for an effect size settled at 0.30 and a level of significance for a *p*-value fixed at <0.05. The third step was to test the effect of treatment on all measures, using paired sample t-test. Finally, to test whether changes in inattention and hyperactivity symptoms were associated with changes in children’s AE and CE, we ran four linear regression models. We used the residualized change, which is the difference between the observed score at postintervention and the predicted score at postintervention, in a two-wave model to measure the change in children’s inattention, hyperactivity symptoms, and both AE and CE [39].

To evaluate changes in the AI, HI, and BES scores, we decided to use the residual changes analysis instead of the simple difference scores (i.e., T1–T0). We preferred the residualized change scores to the difference scores because they adjust for baseline differences and overcome some of the issues with the reliability of different scores. To obtain the residual change scores, first, we performed a series of linear regressions, using the baseline (T0) scores as independent variables and the follow-up (T1) measures as dependent variables. This allowed us to control for differences in the baseline assessment. Unstandardized residuals (i.e., the difference between the T1 observed value and the T1 value predicted by the model; e = y − ŷ) were then saved and used as change scores. The means of all the change scores are equal to zero because they were calculated as residual changes, and the mean of residuals in OLS regressions is always equal to zero. In the first model, we used changes in AE as the dependent variable and changes in inattention symptoms as the independent variable. In the second model, we tested whether changes in hyperactivity symptoms predicted changes in AE. Then, we tested if changes in CE were predicted by changes in inattention or hyperactivity symptoms. In all the regression analyses, we controlled for children’s age and IQ. False discovery rate (FDR; Benjamini and Hochberg, 1995) correction of the *p*-values was applied across all correlations and regression analysis [40]. 

## 3. Results

Descriptive statistics and bivariate correlations between the study variables are shown in Table 1. Results showed that changes in AE were negatively associated with changes in inattention symptoms (*r* = −0.320, *p* < 0.05) and positively associated with changes in CE (*r* = 0.510, *p* < 0.01). The score of the ADHD-RS-Inattention subscale was positively associated with the score of the ADHD-RS-Hyperactivity-Impulsivity subscale at T0 (*r* = 0.51, *p* < 0.01) and at T1 (*r* = 0.64, *p* < 0.01). At the same time, the score of BES-affective subscale was positively associated with the score of BES-cognitive subscale at T0 (*r* = 0.34, *p* < 0.01) and at T1 (*r* = 0.47, *p* < 0.01). During the treatment, the inattention symptoms showed a significant decrease (*t* = 11.970, *p* < 0.01), as well as the hyperactivity/impulsivity symptoms (*t* = 7.412, *p* < 0.01). Besides, cognitive empathy improved (*t* = 5.372, *p* < 0.01), as well as affective empathy (*t* = 6.073, *p* < 0.01).

As regards the linear regression models, the results showed that changes in inattention symptoms did not predict changes in CE (β = −0.194, *p* = 0.143). Furthermore, changes in hyperactivity symptoms did not predict changes in AE (β = −0.148, *p* = 0.258) or CE (β = −0.212, *p* = 0.103). The only significant model was the one with changes in inattention symptoms predicting changes in AE after controlling for children’s age and IQ (see Table 2).

## 4. Discussion

Our study aimed to investigate the effect of MPH on empathic abilities in a group of children and adolescents with ADHD. As expected, after 6 months of MPH treatment, the symptoms of inattention and hyperactivity, as measured at ADHD-RS, showed a significant reduction. At the same time, levels of AE and CE showed a significant increase. We also found that changes in inattention are associated with changes in AE, but not in CE. On the contrary, changes in hyperactivity/impulsivity are not associated with changes in AE or CE, even if we could not exclude the influence of other factors on the change of AE not measured by BES, such as gender or parenting. Our findings suggest that a possible mechanism explaining the social impact of MPH in ADHD youths may be a positive influence of improved attention on AE.

Psychostimulants have proven their efficacy, not only on core symptoms of ADHD, but also on other clinical dimensions affecting the quality of life in a portion of young patients, such as emotional dysregulation and social abilities [41], and have been likely associated with improvements in social judgment and interpersonal relationships [42]. The improvement of social abilities after stimulant treatment may be associated with the decreased intensity of the core symptoms and of their negative effect on social interactions. It has also been suggested that MPH treatment may possibly promote an improvement in emotion recognition [22,43,44], involved in AE [10], by regulating neural activity [2] and, more generally, in empathy [23,30,31]. Concerning that, dopaminergic circuits, a target of pharmacological treatments to ADHD, play a central role in social cognition [32,45].

Our study indicated that 6 months of MPH treatment was associated with a significant improvement in the affective and cognitive empathic ability, as measured with BES, in a group of children and adolescents with ADHD. To our knowledge, this is the first study to have found a significant improvement in both components of empathy after an MPH mid-term treatment and further to have found a positive correlation between the improvement of inattention symptoms and changes in AE. Previously, Golubchik and Weizman (2017) [30] found that a 12-week MPH treatment significantly improved the empathic abilities of young patients with ADHD, although the self-report questionnaire used in this research failed to differentiate the two components of empathy. Gumustas et al. (2017) [31] found that a 12-week MPH treatment was associated with some improvements in situational (i.e., state), but not in dispositional (i.e., trait) empathy skills in youths with ADHD. Demirci & Erdogan (2016) [29] found a significant improvement in theory of mind (ToM) abilities, a construct partially overlapping with empathy [10], in a group of drug-naive young patients with ADHD, about half treated with MPH and about half treated with atomoxetine (ATM) for 12 weeks. In a first pilot study, Maoz et al. (2014) [23] showed that CE abilities in children with ADHD improved after MPH. In a subsequent study, Maoz et al. (2019) [22] found that MPH treatment in children with ADHD improved CE, but not AE, as measured with a self-report questionnaire. In a recent study, Levi-Shachar et al. (2020) examined the effect of a single dose of MPH/placebo on ToM and salivary oxytocin levels, a neuropeptide which regulates social behavior, in a group of children with ADHD and in a group of healthy controls. Following MPH administration, the ToM performance of the children with ADHD, initially poorer, normalized and differences between the two groups were no longer found.

In a meta-analysis, Bora and Pantelis (2016) [46] found that impairment in social cognition in individuals with ADHD lies intermediately between ASD and healthy controls, even if developmental trajectories differ between ADHD and ASD, as social cognitive deficits in ADHD might be improving with age in most individuals. Attentional problems at a very early age have been supposed to precede the onset of clinical manifestations of ASD, ADHD, or both disorders [47]. In this line, the association between ASD and ADHD traits may be due to shared attention-related problems (inattention and attentional switching capacity) and biological pathways involving attentional control may be a key factor in the overlapping conditions [48,49].

Our findings suggest that an effective therapeutic intervention on attention in ADHD patients may be associated with an improvement of social cognition and, in particular, of AE. Future studies could explore the effects of MPH on AE also in patients with co-occurring ASD. Some studies demonstrated the positive correlation between empathic competences and executive functions (EFs) in healthy subjects [50] and in clinical samples [42,51]. In a recent meta-analysis, Yan et al. (2020) [50] found that EFs were more strongly related to CE. However, the Authors also indicated that AE was closely related to inhibitory control. Conversely, the paper by Cristofani et al. [51], which assessed the reciprocal relationship between empathic attitudes and EFs in ADHD patients, indicated that EFs were more strongly related to the AE than to the cognitive one. The Authors speculated that ADHD patients are somewhat constrained by their executive disfunction in an underdevelopment of their empathic attitude, which would be limited to the expression of an emotional contagion. In an other meta-analysis, Tamminga et al. (2016) [52] examined the effects of MPH on executive functions in children, youths and adults with ADHD and found that the effects on response inhibition, working memory and sustained attention were small to moderate. We speculate that the improvement of inattention symptoms, observed in the current sample, is related to the inhibitory control, which, in turn, is involved in the improvement of AE. The positive effect of MPH on EFs may be explored as a possible further mediator in the improvement of empathy and social abilities in youths with ADHD and/or ASD.

At the pharmacological level, the first effect of MPH is related to increased central dopaminergic and noradrenergic activity in the brain regions that include the cortex and striatum, regions involved in the regulation of executive and attentional functions [53]. Interestingly, several studies demonstrated that empathic attitudes are activated through an emotional processing which is regulated both by bottom-up and top-down circuitry within the prefrontal and limbic cortex [50].

Our preliminary findings must be viewed in light of several limitations, first, the relatively small sample size. Second, we did not have a healthy control group, or a control group receiving a placebo in a double-blind design. Third, the information obtained from the BES questionnaires presents the limits of all the self-report measures, and it should be integrated with parent reports or experimental paradigms. Fourth, no measures were used to find the socioeconomic level of participants and it should be considered in future studies. Finally, the present study does not consider the effects on empathy of other variables, such as gender [54] or parenting [55] and cognitive process [44].

## 5. Conclusions

In conclusion, our study provides a further contribution for a better understanding of the possible effects of the MPH. This evidence supports the notion that the timely and affective treatment of ADHD symptoms may have beneficial effects not only on core symptoms of ADHD, but also on the social difficulties of youths with ADHD. Furthermore, our findings suggest that attention may be a primary target of a pharmacological or psychological or psychoeducational intervention that aims to improve AE in youths with ADHD. Future studies on the association of several measures of empathy with comorbid disorders, such as ASD and disruptive behavioral problems, are warranted. At the same time, considering the absence of a healthy control group, we could not exclude the possibility that the improvement of AE in our patients could be due to learning; about this, a placebo-controlled study might be useful.

## Figures and Tables

**Table 1 children-08-00596-t001:** Correlations, means and standard deviations of the study variables.

	1	2	3	4	5	6	7	8	9	10	11	12	13	14
1. Age	1													
2. IQ	0.01	1												
3. ADHD-RS Inatt. 1	−0.04	0.03	1											
4. ADHD-RS Hyperact. 1	−0.39 **	0.01	0.51 **	1										
5. BES affective 1	0.16	−0.01	−0.40 **	−0.26 *	1									
6. BES cognitive 1	−0.10	0.06	−0.23	−0.19	0.34 **	1								
7. ADHD-RS Inatt. 2	0.09	0.14	0.54 **	0.24	−0.17	−0.20	1							
8. ADHD-RS Hyperact. 2	−0.17	0.07	0.39 **	0.61 **	−0.12	−0.14	0.64 **	1						
9. BES affective 2	0.10	−0.17	0.01	0.09	0.30 *	−0.15	−0.27 *	−0.06	1					
10. BES cognitive 2	−0.17	−0.01	−0.02	0.05	−0.02	0.00	−0.19	−0.14	0.47 **	1				
11. Change Inattention	0.13	0.14	0.00	−0.04	0.06	−0.09	0.83 **	0.51 **	−0.25 *	−0.21	1			
12. Change Hyperactivity	0.08	0.07	0.10	0.00	−0.02	−0.02	0.63 **	0.78 **	−0.15	−0.22	0.68 **	1		
13. Change BES affective	0.06	−0.17	0.14	0.18	0.00	−0.26 *	−0.16	−0.01	0.95 **	0.50 **	−0.32 *	−0.15	1	
14. Change BES cognitive	−0.16	−0.01	−0.02	0.05	−0.02	0.00	−0.18	−0.14	0.48 **	0.80 **	−0.21	−0.25	0.51 **	1
Mean	10.29	89.52	18.67	15.20	29.73	31.06	11.37	9.87	32.75	33.52	0	0	0	0
SD	2.84	10.43	4.90	6.77	4.98	4.19	5.08	5.92	3.39	3.00	4.26	4.66	3.23	3.00

IQ = intelligence quotient; ADHD-RS Inatt. 1 = ADHD-Rating Scale-Inattention subscale (T0); ADHD-RS Hyperact. 1 = ADHD-Rating Scale-Hyperactivity-Impulsivity subscale (T0); ADHD-RS Inatt. 2 = ADHD-Rating Scale-Inattention subscale (T1); ADHD-RS Hyperact. 2 = ADHD-Rating Scale-Hyperactivity-Impulsivity subscale (T1); BES affective 1 = Basic Empathy Scale-affective subscale (T0); BES cognitive 1 = Basic Empathy Scale-cognitive subscale (T0); BES affective 2 = Basic Empathy Scale-affective subscale (T1); BES cognitive 2 = Basic Empathy Scale-cognitive subscale (T1); Change Inattention = change in ADHD-RS Inattention subscale’s score after the six months of MPH treatment; Change Hyperactivity = change in the ADHD-RS Hyperactivity-Impulsivity subscale’s score after the six months of MPH treatment; Change BES affective = change in BES-affective subscale’s score after the six months of MPH treatment; Change BES cognitive = change in BES-cognitive subscale’s score after the six months of MPH treatment; Mean = mean of the above variables; SD = standard deviation of the above variable; SD = standard deviation. The means of all the change scores are equal to zero because they were calculated as residual changes, and the mean of residuals in OLS regressions is always equal to zero. * *p* < 0.05; ** *p* < 0.01.

**Table 2 children-08-00596-t002:** Multivariate regression where changes in inattention measured by ADHD-RS predicted the change in children’s affective empathy measured by BES.

	B	SE	β
IQ	−0.04	−0.04	−0.13
Age	0.12	0.14	0.10
Change Inattention	−0.22	0.97	−0.28 *
R^2^	0.12		

R^2^ = coefficient of determination; * *p* = 0.02.

## Data Availability

The data presented in this study are available on request from the corresponding author.

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
