# Peer review of "Treatment with Methylphenidate Improves Affective but Not Cognitive Empathy in Youths with Attention-Deficit/Hyperactivity Disorder"

_children, 2021, doi:10.3390/children8070596_

Round 1
Reviewer 1 Report
Given the high comorbid rate of ASD with ADHD, why authors decided to include ASD as exclusion criteria? I can speculate the reason, but authors should provide readers with the rationale for it. In other words, authors may inform readers of the close relationship between ASD and empathy (deficit).
Can the K-SADS-PL diagnose language disorder, verbal dyspraxia, or ASD? If not, how authors diagnosed them.
In the discussion, authors suggested the possible reason for the improvement of AE via an effective therapeutic intervention on attention in ADHD patients. They referred Yan’s finding that affective empathy is closely related to inhibitory control. Authors should explain this in more detail since in general, the inhibitory control is more tightly connected with impulsivity than inattention.
Author Response
Given the high comorbid rate of ASD with ADHD, why authors decided to include ASD as exclusion criteria? I can speculate the reason, but authors should provide readers with the rationale for it. In other words, authors may inform readers of the close relationship between ASD and empathy (deficit).
As you suggested, a wide range of studies reported a lack of empathy in ASD. We modified the introduction based on your suggestion, see line 48-51. Considering that ASD is a neurodevelopmental disorder often associated with a lack of empathy, we decided to exclude patients with ASD comorbidity in this pilot study (line 84-86).
Can the K-SADS-PL diagnose language disorder, verbal dyspraxia, or ASD? If not, how authors diagnosed them.
K-SADS-PL can not diagnose language disorder, verbal dyspraxia, or ASD. The two patients with Language Disorder and the patient with Verbal Dyspraxia had previously received these diagnoses in the Neurology section of our Department. The ASD diagnosis were given by experienced clinicians according to the criteria of the DSM-5 (American Psychiatric Association, 2013) and it was made with Autism Diagnostic Observation Schedule, Second Edition (ADOS-2; Lord et al., 2012).
We modified the description of the group based on your suggestion, see line 101-105.
In the discussion, authors suggested the possible reason for the improvement of AE via an effective therapeutic intervention on attention in ADHD patients. They referred Yan’s finding that affective empathy is closely related to inhibitory control. Authors should explain this in more detail since in general, the inhibitory control is more tightly connected with impulsivity than inattention.
We modified the discussion based on your suggestion, see line 262-269. We explained that Yan et al. (2020) found that EFs were more strongly related to CE. However, the Authors also indicated that AE was closely related to inhibitory control. Conversely, the paper by Cristofani et al., which assessed the reciprocal relationship between empathic attitudes and EFs in ADHD patients, indicated that EFs were more strongly related to the AE than to the cognitive one. The Authors speculated that ADHD patients are somewhat constrained by their executive disfunction in an underdevelopment of their empathic attitude, which would be limited to the expression of an emotional contagion.
Reviewer 2 Report
1. It's a great study to evaluate effect of pharmacologic treatment on social impairment of ADHD, with sufficent follow up on treatment naive patients.
2. Is there supportive evidence or theory from literature can explain AE but not CE were correlated with improvement of inattention symptoms ?
3. The authors explored four regression models with only one model revealed significant correlation. An possible effect of multiple hypothesis tests should be addressed.
Author Response
It's a great study to evaluate effect of pharmacologic treatment on social impairment of ADHD, with sufficent follow up on treatment naive patients.
Tank you for appreciating our study.
Is there supportive evidence or theory from literature can explain AE but not CE were correlated with improvement of inattention symptoms?
To our knowledge, this is the first study to found a significant improvement of both components of empathy after an MPH mid-term treatment and to further found a positive correlation between improvement of inattention symptoms and changes in AE.
At the pharmacological level, the first effect of MPH is related to increased central dopaminergic and noradrenergic activity in brain regions that include the cortex and striatum, regions involved in the regulation of executive and attentional functions. Interestingly, several studies demonstrated that empathic attitudes are activated through an emotional processing which is regulated both by bottom-up and top-down circuitry within the prefrontal and limbic cortex.
We modified the discussion based on your suggestion.
The authors explored four regression models with only one model revealed significant correlation. An possible effect of multiple hypothesis tests should be addressed.
False discovery rate (FDR) correction of the p‐values was applied across all correlations and regression analysis. See line 167-169.
Round 2
Reviewer 2 Report
I have no further comments.